# Zero-Shot Model Search via Text-to-Logit Matching

## Abstract

With the increasing number of publicly available models, there are pre-trained, online models for many tasks that users require. In practice, users cannot find the relevant models as current search methods are text-based using the documentation which most models lack of. This paper presents ProbeLog, a method for retrieving classification models that can recognize a target concept, such as "Dog", without access to model metadata or training data. Specifically, ProbeLog computes a descriptor for each output dimension (logit) of each model, by observing its responses to a fixed set of inputs (probes). Similarly, we compute how the target concept is related to each probe. By measuring the distance between the probe responses of logits and concepts, we can identify logits that recognize the target concept. This enables zero-shot, text-based model retrieval ("find all logits corresponding to dogs"). To prevent hubbing, we calibrate the distances of each logit, according to other closely related concepts. We demonstrate that ProbeLog achieves high retrieval accuracy, both in ImageNet and real-world fine-grained search tasks, while being scalable to full-size repositories. Importantly, further analysis reveals that the retrieval order is highly correlated with model and logit accuracies, thus allowing ProbeLog to find suitable and accurate models for users tasks in a zero-shot manner.

## 1 Introduction

Neural networks have revolutionized fields such as computer vision (He et al., 2016; Dosovitskiy, 2020; Redmon, 2016; Li et al., 2023; Rombach et al., 2022) and natural language processing (Touvron et al., 2023; Devlin, 2018; Vaswani, 2017), becoming indispensable tools for many real-world classification tasks. However, their high training cost leaves users with two suboptimal options: i) invest heavily in computational resources for training or fine-tuning a model, ii) settle for a general-purpose model which with substantial inference cost. Now, imagine that instead, one could simply search online for the most accurate model for their specific task and use it directly without additional training. With the rise of large public model repositories, this is becoming feasible. For instance, Hugging Face, the largest existing model repository, hosts over a million models, with more than $100,000$ models added each month. This significantly increases the likelihood of finding a suitable public model for most user tasks. However, the main challenge lies in retrieving the right model for each task. Current model search methods (Shen et al., 2024; Luo et al., 2024) rely on provided metadata or text descriptions, while in practice most models are either undocumented or have very limited descriptions (See Fig. 1), which severely limits these methods ability to retrieve suitable models.

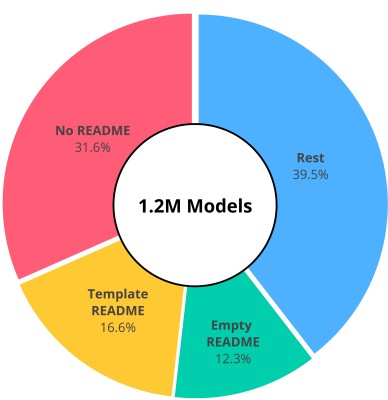

Figure 1: **HuggingFace Documentation.** We analyze over $1M$ model cards from Hugging Face, showing that most models are either undocumented or poorly documented.

We aim to search for new models based on their weights, without assuming access to their training data or metadata, as these are often unavailable. More precisely, the goal is to retrieve all classification models capable of recognizing a particular concept, such as "Dog". For a solution to be

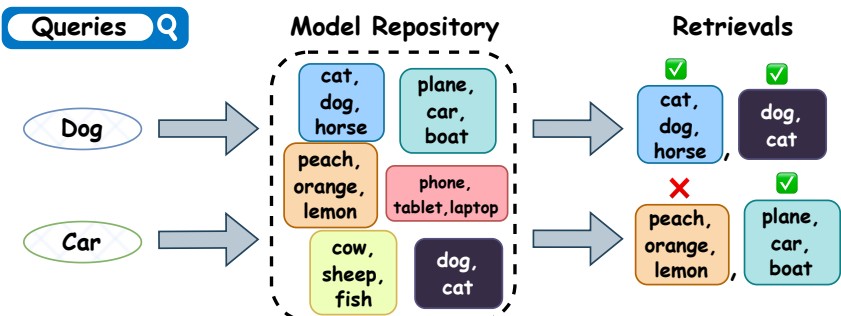

Figure 2: *Classification Model Search.* We present a new task of Classification Model Search, where the goal is to find classifiers that can recognize a target concept. Concretely, given an input prompt, such as "Dog", we wish to retrieve all classifiers that one of their classes is "Dog". The search space is a large model repository, which contains many models and concepts to search from. The retrieved models can replace model training, increase accuracy and reduce computational cost.

effective and practical, it must meet several requirements: i) identify models that recognize the target concept, regardless of the other concepts they can detect and their order, ii) scale to large model repositories, ii) support text-based search, and iv) retrieve the most accurate models. This task is challenging, as it requires understanding what neurons do. While previous approaches (Oikarinen & Weng, 2023; Bau et al., 2017) have made great progress in zero-shot neuron concept classification, they were not designed for text-based model search.

In this paper, we present *ProbeLog*, a method designed for the new task of Classification Model Search. We begin by analyzing the performance of zero-shot neuron classification methods on model retrieval (search). We show these methods suffer from *hubbing*, where many different queries retrieve only a small fraction of the concept gallery. We therefore propose a simpler approach. Given a set of query probes and a concept name we wish to look for, we compute the i) logit response for each probe and ii) CLIP's cosine similarity between the concept name and each probe. We find that a truncated euclidean distance between logit responses and concept-probe similarities (via CLIP) provides an effective matching metric, while being considerably simpler than the pointwise mutual information used by state-of-the-art zero-shot neuron classification methods. However, in the case of model retrieval, the hubness issue remains: some logits are much closer to many concepts. Therefore, our final method, ProbeLog, proposes a hubness correction term which overcomes this issue. Lastly, we demonstrate that ProbeLog is especially practical for model search, as its retrieval order is highly correlated with the accuracies of the models. Thus, by taking ProbeLog's first retrieval, users can find the most relevant and accurate model for their task.

We showcase ProbeLog's effectiveness on two real-world datasets that we curate: one based on models that we train on ImageNet subsets and the other containing models that we download from Hugging Face. Our method is scalable and can handle large models with high effectiveness and efficiency. It achieves high retrieval accuracy, reaching $42.6\%$ top-1 retrieval accuracy when predicting whether an in-the-wild model can recognize a target concept from text.

Our main contributions are:

1. Reevaluating zero-shot logit classification and showing a simple truncated euclidean distance approach compares favorably to the state-of-the-art.

2. Introducing ProbeLog, a method for text-based model search.

3. Introducing 2 new model zoos for this task (including 1300 real model logits from HuggingFace).

## 2 RELATED WORKS

### 2.1 WEIGHT-SPACE LEARNING

While neural networks can learn effective representations for many traditional data modalities, effective representations for neural networks are still a work in progress. Unterthiner et al. (2020) took

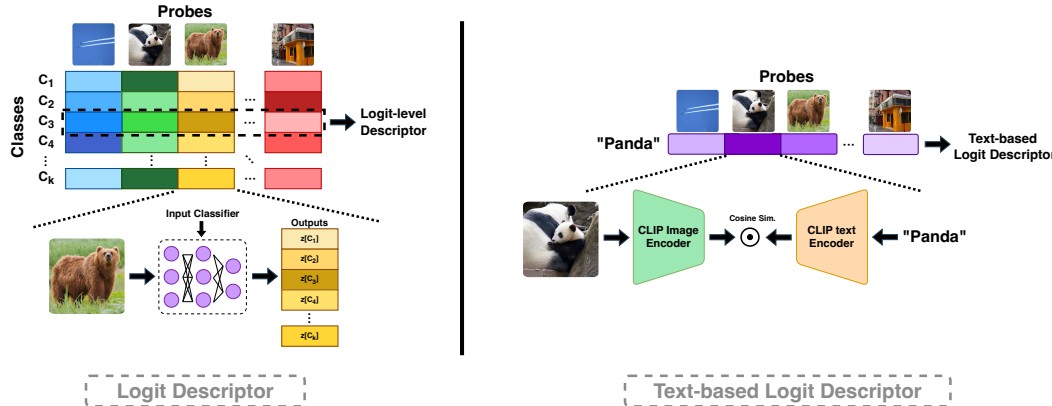

Figure 3: **ProbeLog Descriptors.** Our method generates descriptors for individual output dimensions (logits) of models. First, we sample a set of inputs (e.g., from the COCO dataset), and fix them as our set of probes. Then, to create a new logit descriptor, we feed the set of ordered probes into the model and observe the responses on the matching output dimension. Similarly to Oikarinen & Weng (2023), we extend this idea to zero-shot concept descriptors using CLIP, embedding text into the logit descriptors space as the vector of similarities between the text and our set of probes according to CLIP. Lastly, we find that by normalizing both descriptors we can compare them directly.

the first step, proposing to observe simple statistics of weights, and use (Ke et al., 2017) on them. Others proposed encoding the weights by modeling the connections between neurons (Navon et al., 2023; De Luigi et al., 2023; Schürholt et al., 2024; 2021; Eilertsen et al., 2020; Lim et al., 2024; Zhou et al., 2024a; Tran et al., 2024; Dupont et al., 2022; Horwitz et al., 2024a). Recent methods (Kofinas et al., 2024; Zhou et al., 2024b; Lim et al., 2023; Kalogeropoulos et al., 2024) model a network as a graph where every neuron is a node, and train permutation-equivariant architectures (Gilmer et al., 2017; Kipf & Welling, 2016; Diao & Loynd, 2022) on these graphs. Probing is an alternative paradigm that encodes the network by observing its outputs on a fixed set of inputs (probes) (Kahana et al., 2024; Herrmann et al., 2024; Carlini et al., 2024; Tahan et al., 2024; Choshen et al., 2022; Kofinas et al., 2024; Huang et al., 2024). Learning on model weights has found many applications including advanced generation abilities (Dravid et al., 2024; Erkoç et al., 2023; Dravid et al., 2024; Shah et al., 2023), model compression (Ha et al., 2016; Ashkenazi et al., 2022; Peebles et al., 2022), model graph recovery (Horwitz et al., 2025; 2024c; Yax et al., 2024), model merging (Yadav et al., 2024; Gueta et al., 2023; Izmailov et al., 2018; Wortsman et al., 2022; Ramé et al., 2023), and even recovering black-box models (Horwitz et al., 2024b; Carlini et al., 2024). Some relevant works search for new adapters for generative models (Shen et al., 2024; Luo et al., 2024; Lu et al., 2023), however these approaches either rely on available metadata or tailored for generative models.

## 2.2 INTERPRETING INDIVIDUAL NEURONS

Several works attempted to describe the function of individual neurons by visualizations of relevant inputs (Zeiler & Fergus, 2014; Girshick et al., 2014; Mahendran & Vedaldi, 2015; Karpathy et al., 2015). Others use automatic categorization to classify each neuron to a known set of classes (Bau et al., 2017; 2020; Oikarinen & Weng, 2023; Dalvi et al., 2019) or choose natural language to describe neurons (Schwettmann et al., 2021; Hernandez et al., 2021a; Gandelsman et al., 2023; Shaham et al., 2024). In this work, we tackle the problem of model search, where, given a task of choice, one searches for a suitable model for the task inside a model gallery. We show that neuron classification is highly connected with model search and propose how to successfully adapt previous methods of neuron classification (Oikarinen & Weng, 2023) for retrieval of suitable models.

## 3 BACKGROUND AND MOTIVATION

### 3.1 PROBLEM DEFINITION: MODEL SEARCH

We assume a model repository composed of $m$ classifiers, $f_1, f_2, ..., f_m$. Each classifier $f_i$ can have multiple output dimensions (logits), each corresponding to an unknown concept $l_{i,j}$. The user then

Table 1: ***Neuron Classification Results.*** We evaluate the Top-1 and Top-5 neuron classification accuracies of our method and CLIP-Dissection on the INet-Hub (synthetic) and HF-Hub (real world). As both methods are comparable, we conclude our simplified approach is indeed at least as good.

| | Top-1 Accuracy | | Top-5 Accuracy | |
| --- | --- | --- | --- | --- |
| Method | text $\rightarrow$ INet | text $\rightarrow$ HF | text $\rightarrow$ INet | text $\rightarrow$ HF |
| CLIP-Dissect (WPMI) | $26.0\%_{\pm 0.4}$ | $44.7\%_{\pm 0.4}$ | $55.2\%_{\pm 0.4}$ | $66.2\%_{\pm 0.7}$ |
| CLIP-Dissect (SoftWPMI) | $14.7\%_{\pm 0.1}$ | $37.1\%_{\pm 1.1}$ | $31.7\%_{\pm 0.4}$ | $54.7\%_{\pm 1.0}$ |
| All probes + Anti-Hub | $13.6\%_{\pm 0.1}$ | $22.0\%_{\pm 0.4}$ | $31.6\%_{\pm 0.6}$ | $37.4\%_{\pm 0.7}$ |
| ProbeLog (Ours) | $26.2\%_{\pm 0.2}$ | $45.3\%_{\pm 0.7}$ | $52.1\%_{\pm 0.4}$ | $67.2\%_{\pm 0.2}$ |

inputs a text prompt containing some query concept, $c$, they wish to search for. Finally, the goal is to return a model $f_i$ such that one of its classes matches the query concept. Formally, the set of all valid retrieval models, $R(c)$, is defined as:

$$R(c) = \{f_i \mid \exists j \; s.t. \; l_{i,j} = c\} \tag{1}$$

As mentioned above, the retrieval algorithm does not know the class concepts of each model. We assume access to them solely for evaluation purposes.

## 3.2 THE CHALLENGE: REAL MODELS ARE POORLY DOCUMENTED

The existing solution for model search is text-based search in the user-uploaded documentation. To understand the effectiveness of this solution, we explore the level of documentation of models in Hugging Face, the largest model repository. For that, we analyzed $1.2M$ model cards. As shown in Fig. 1, over $30\%$ of all models have no model card at all. Moreover, there are another $28.9\%$ of model cards that are either empty or include an empty automatic template with no information. The remaining $40\%$ of model cards may include some information, however we cannot determine exactly how many of them include relevant information about the training data. As most models are poorly documented, it is important to look for alternative search methods. As all models with API access can be probes, we develop a probing-based approach to model search.

## 3.3 NETWORK DISSECTION METHODS STRUGGLE ON SEARCH

Network dissection methods (Oikarinen & Weng, 2023; Bau et al., 2017; Shaham et al., 2024; Hernandez et al., 2021b) aim to classify the concepts of monosemantic neurons. They first probe the model with a set of curated samples (probes) and identify the probes that highly activate each neuron. Each neuron is then labelled by the main repeating concept across the set of highly activated probes. CLIP-Dissect (Oikarinen & Weng, 2023) proposed zero-shot neuron classification, by choosing the target concept whose mostly activated images have the highest mutual information according to CLIP (Radford et al., 2021). As a first step, we test CLIP-Dissect for model search on a real-world Model Zoo (Schürholt et al., 2022) collected from HuggingFace (see App. F) with 1300 logits spanning more than 500 different concepts. Results are presented in Tab. 1. While CLIP-Dissect reaches decent performance of $35.3\%$ top-1 accuracy, Fig. 4 shows it suffers from severe hubness. I.e., many of the top retrievals come from the same small set of logits.

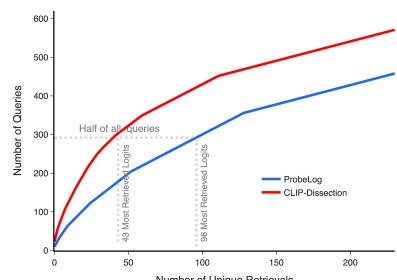

Figure 4: ***Hubness in CLIP-Dissect and ProbeLog.*** Hubness causes many queries to retrieve a handful of unique logits. Here, we show the number of queries covered by the top K logits for both methods. We observe that ProbeLog has significantly fewer hub logits.

## 4 METHOD

In this section, we first present a zero-shot classification method for logits by directly comparing texts and logits. It has similar accuracy as other popular approaches (Oikarinen & Weng, 2023) while being much simpler. We then show how to adapt this method for model search.

## 4.1 CONCEPT-LOGIT TRUNCATED DISTANCE

Our objective is to accurately and efficiently find relevant models in a large repository that can recognize a target concept, e.g., "Husky Dog". To do so, we probe the model with several input images and record the logit response for each one. Formally, we input each probe $x$ into the model $f$, obtaining the response $f(x)[j]$ in the model's $j^{th}$ logit. We denote the response at a logit $l$ for the $j^t h$ output neural of model $i$ as the responses of all probes at this logit:

$$\phi_{logit}(l) = [f_i(x_1)[j], f_i(x_2)[j], \cdots, f_i(x_n)[j]] \tag{2}$$

We also compute the similarity between the concept and each of the probes. We do so using CLIP, by using its text encoder to embed the concept text and its image and encoder to embed the image. The cosine similarity of these embeddings provides the similarity between the concept and probe:

$$\phi_{text}(c) = [CLIP(x_1, c), CLIP(x_2, c), \cdots, CLIP(x_n, c)] \tag{3}$$

We illustrate the creation of our zero-shot text-based logit descriptors in Fig. 3. To compare a logit $l$ and text concept $c$ we can simply measure the euclidean distance of the probe responses:

$$d(l, c) = \ell_2(\phi_{logit}(l), \phi_{text}(c)) \tag{4}$$

However, there are two issues with using the euclidean distance. First, probe similarities of logits and CLIP might have very different scales, and must be normalized to compare properly. Secondly, we find that probes which the classifier is not confident about, i.e., probes for which logit responses are low, provide much less information and harm retrieval performance. To combat this, we ignore all probes that have low response values, keeping only the top $r$ probes (in our experiments we choose $r = 50$). Formally, let $a = [a_1, a_2, \ldots, a_n]$ be the sorted probe indices in descending order according to $\phi_{logit}(l)$, our distance measure becomes:

$$d_{trim}(l, c) = \sqrt{\sum_{i=1}^{r} \left(\frac{\phi_{logit}(l)[a_i] - \mu_l}{\sigma_l} - \frac{\phi_{text}(c)[a_i] - \mu_c}{\sigma_c}\right)^2} \tag{5}$$

where $\mu_l, mu_c$ and $\sigma_l, sigma_c$ are the mean and standard deviation on the entire logit and concept descriptor (not only the top probes). We call this measure: *truncated euclidean distance*. We compare our simple approach with the more complex pointwise mutual information approaches of CLIP-Dissect, for neuron classification accuracy. Results are presented in Tab. 1. Our truncated distance compares favorably with PMI while being linear and not requiring probabilistic estimation.

## 4.2 HUBNESS CALIBRATION

In Sec. 3.2 we showed that a standard neuron classification approach suffers from hubness, where most queries return the same logits (hubs), although more suitable logits exists. To mitigate that, we propose to calibrate the hubness, essentially down weight hubs. This is inspired by techniques from retrieval (Lample et al., 2018). However, at inference time we receive one query at a time, meaning we cannot detect hubs as we see only a single distance from each logit. Therefore, we create a background set of concepts. Specifically, we randomly choose 500 classes from ImageNet-21K (Deng et al., 2009), and use these to calibrate the distances of each logit. Given a query descriptor $\phi_c$ and gallery logit descriptor, $\phi_l$, we compute the truncated distances of $\phi_l$ to all 500 chosen class descriptors $\phi_{c_1}, ..., \phi_{c_{500}}$. We then subtract the mean of the $k$ smallest distances from $d_{trim}(\phi'_c, \phi_c)$. Formally, let $b = [b_1, b_2, \ldots, b_n]$ be the indices of the sorted truncated distances in descending order. Our calibration then becomes:

$$d_{trim}(l, c) \leftarrow d_{trim}(l, c) - \frac{1}{k} \sum_{i=1}^{k} d_{trim}(l, c_{b_i}) \tag{6}$$

Table 2: ***Retrieval Results.*** We evaluate the Top-1 and Top-5 retrieval accuracies of our method and the baselines for text-based retrievals and logit classification. All methods use COCO images as probes. For a fair comparison, all experiments are performed with $4,000$ probes.

| | Top-1 Accuracy | | Top-5 Accuracy | |
|---|---|---|---|---|
| Method | text $\to$ INet | text $\to$ HF | text $\to$ INet | text $\to$ HF |
| CLIP-Dissect (WPMI) | $44.9\%_{\pm 0.4}$ | $35.3\%_{\pm 1.2}$ | $67.1\%_{\pm 0.8}$ | $50.7\%_{\pm 1.6}$ |
| CLIP-Dissect (SoftWPMI) | $42.1\%_{\pm 0.6}$ | $35.1\%_{\pm 1.3}$ | $64.1\%_{\pm 0.5}$ | $49.2\%_{\pm 0.9}$ |
| All probes + Anti-Hub | $58.7\%_{\pm 1.3}$ | $31.8\%_{\pm 1.2}$ | $78.9\%_{\pm 0.7}$ | $49.4\%_{\pm 0.3}$ |
| **ProbeLog (Ours)** | $\mathbf{64.4\%}_{\pm 0.4}$ | $\mathbf{42.6\%}_{\pm 1.1}$ | $\mathbf{82.5\%}_{\pm 0.7}$ | $\mathbf{60.1\%}_{\pm 0.9}$ |

Intuitively this means that if the logit represents the class "Cat" we ask whether this logit is more similar to "Cat" than "Tiger", "Puma" or "Lynx", rather than just asking whether this is a "Cat" against all classes, which could aid in detecting fine-grained concepts.

### 4.3 RELATION TO CURRENT CONCEPT-LOGIT SIMILARITY MEASURES

In this section, we show that our truncated distance strikes a good balance between previous linear and non-linear similarity measures. Previously proposed linear measures used the inner product of the concept and logit descriptor, however this was shown (Oikarinen & Weng, 2023) to fail. WPMI works much better but is much more complex:

$$\text{wpmi}(c, l) = \sum_{i=1}^{r} \log p(c|x_{a_i}) - \lambda \log \sum_{a' \in A} \left( \Pi_{i=1}^{r} p(c|x_{a'_i}) \right) + \lambda \log |C| \qquad (7)$$

Computing this quantity requires estimating the probability of the concept given the probe, which is difficult to do precisely. It also assumes independence of between the probes which might not always be true. Additionally, its normalization term is quite complex and includes a combinatorial sum of subparts. The soft weighted PMI is even more complex and is detailed in Appendix A of (Oikarinen & Weng, 2023). In contrast, our truncated distance method is much simpler and does not suffer from the above limitations. It also vastly improves over simple inner products by: i) normalizing the logit and concept features, so they operate in the same scale, and ii) computing the inner product on just the top $r$ probes. Our method therefore strikes a better balance between simplicity and performance.

## 5 EXPERIMENTS

### 5.1 EXPERIMENTAL SETTING

**Datasets.** As there are no suitable existing datasets for model search that include ground-truth data, we created 2 new ones, INet-Hub and HF-Hub. For each model in the INet-Hub (see App. E), we sample a subset of ImageNet classes, a model architecture and foundation model initialization checkpoint. We then train the model on the selected data. The final dataset consists of $1,500$ models, making a total of more than $85,000$ logits, derived from $1000$ unique fine-grained concepts (see App. E). Our second hub, HF-Hub, is a set of $1300$ real-world model logits (collected from over $250$ different models) downloaded from HuggingFace (see App. F).

**Baselines.** We test our retrieval algorithm against two baselines: (i) CLIP-Dissect (Oikarinen & Weng, 2023), which is designed for neuron classification, and measures how confident is CLIP on the mostly activated probes of each neuron. WPMI and SoftWPMI are two variants of CLIP-Dissect different in their weighting of different probes in the mutual information calculation. (ii) normalized euclidean distance using all probes, instead of truncating the top-$r$ probes as in our final approach. For this baseline we also include our anti-hubness calibration.

**Metrics.** We evaluate the retrieval performance using the standard top-k accuracy (with $k \in [1,5]$). Top-k accuracy measures the percentage of target logits that had a relevant result in any of their top-k retrieved logits.

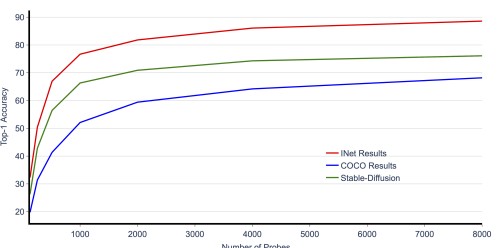

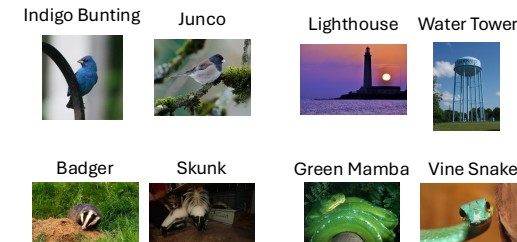

Figure 5: ***Number of Probes.*** We test ProbeLog on the INet-Hub with increasing numbers of probes. While more probes lead to higher accuracy, the gains are diminishing.

Figure 6: ***ProbeLog Semantic False Retrievals.*** We visualize 4 random retrieval errors and see the errors are generally semantic.

## 5.2 MODEL SEARCH RESULTS

We evaluate our method on the 2 Model Zoos and present the results in Tab. 2. We report both the top-1 and top-5 accuracies. Here, we evaluate model retrieval performance, i.e., search-by-text evaluation where we search for the closest retrievals to a zero-shot text descriptor in either the INet-Hub or the HF-Hub. We can see that in both cases, our approach greatly exceeds the baselines, reaching an impressive top-1 accuracy of $64.4\%$ on the INet-Hub. Moreover, when tested on the HF-Hub we can see that our method generalizes to real-world models, as it finds suitable matches for more than a $40\%$ of the queries in the first search result, and for more than $60\%$ of queries within the first 5 retrievals. This shows that while simple, our approach can generalize to real-world scenarios. In App. A we provide additional results, showing that our method can also match between images and model logits, allowing to search for concepts which are difficult to describe by text.

## 5.3 ERRORS OR NEAR-MISSES?

Retrieval mistakes can vary in severeness, e.g. given a query concept of "German Shepherd" a retrieval of "Husky Dog" is more forgivable than "Pickup Truck". To qualitatively evaluate the severity of our mistakes we randomly sample a few random wrong retrievals and plot the images matching to their concepts. We visualize 4 such random retrievals in Fig. 6. For more uncurated retrieval examples see App. G. We can see that ProbeLog's mistakes are often near misses rather than random errors. E.g., its mistake for "Green Mamba" is simply another similar snake species.

## 5.4 RETRIEVED MODELS ACCURACY

While ProbeLog shows impressive results for retrieving a logit with the right concept, it is interesting to evaluate how well the returned models recognize the query class. To test that, we evaluate the first correct retrieval for each query on the INet-Hub using precision-recall Area Under Curve (PR-AUC), when tested with the entire test set of ImageNet. We test the retrieved logits in a one-vs-all setting where samples are labeled 1 for the query class and 0 otherwise. Note, this means many of the samples used are out-of-distribution to the model, as each model was trained on a subset of ImageNet classes. We compare the PR-AUC of ProbeLog retrieved models against the zero-shot PR-AUC obtained by CLIP in the same setting. I.e., we score the ImageNet test set according to the text prompt description of the class in question, and compute the PR-AUC of CLIP itself. Although large language models might score better than CLIP in this setting, we do not include them in our comparison, as they are much more computationally demanding, having $1000\times$ as many parameters as the specialized classification models. The results are presented in Tab. 3. We can clearly see that ProbeLog finds models that are much more accurate than the average model available on the INet-Hub as well as CLIP itself, by a large margin. Moreover, to test whether ProbeLog ranks accurate models higher, we present the average PR-AUC over the retrieval ranks in Fig. 7. We can see that indeed better models are returned first, demonstrating that to use ProbeLog, one can simply take the its top-1 retrieval. In App. B we provide a similar analysis on model accuracy rather than logit accuracy, which shows that ProbeLog's top retrieved logits arrive from highly accurate models (surpassing CLIP by up to $8.9\%$ on average).

Table 3: ***PR-AUC of Retrieved Logits.*** ProbeLog's top retrievals are much better than both the average model in the repository and CLIP zero-shot binary classification.

|  | PR-AUC |
|---|---|
| Mean INet-Hub | 0.433 |
| CLIP Zero-shot | 0.421 |
| **ProbeLog** | **0.647** |

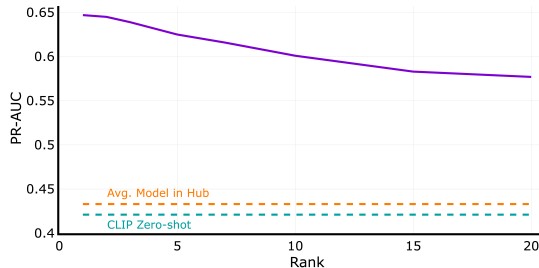

Figure 7: ***PR-AUC vs. Retrieval Rank.*** ProbeLog retrieves accurate logits first. For model-level accuracy see App. B.

Table 4: ***Dataset Ablations.*** We compare both real and synthetic probe distributions. While distributions closer to the model's training data lead to better results, even out-of-distribution probes sampled from the COCO dataset retrieve relevant logits with high accuracy.

| Method | Top-1 Accuracy | | Top-5 Accuracy | |
|---|---|---|---|---|
| | text → INet | text → HF | text → INet | text → HF |
| Dead-Leaves | $4.1\%_{\pm0.3}$ | $2.8\%_{\pm0.3}$ | $11.4\%_{\pm0.5}$ | $8.2\%_{\pm0.6}$ |
| Stable-Diffusion | $73.9\%_{\pm0.6}$ | $54.2\%_{\pm0.9}$ | $89.9\%_{\pm0.7}$ | $72.8\%_{\pm0.7}$ |
| ImageNet | $86.1\%_{\pm0.6}$ | $57.3\%_{\pm0.5}$ | $95.3\%_{\pm0.3}$ | $75.3\%_{\pm1.4}$ |
| COCO | $64.4\%_{\pm0.4}$ | $42.6\%_{\pm1.1}$ | $82.5\%_{\pm0.7}$ | $60.1\%_{\pm0.9}$ |

## 5.5 HUBBING ANALYSIS

We evaluate ProbeLog's hubness similarly to the evaluation in Sec. 3.2, checking whether a small set of logits are still responsible for the most returns. We provide the results in Fig. 4. We see that indeed hubness is substantially reduced, and more logits are included in the top-1 returns. Specifically, while in CLIP-Dissection the 43 biggest hubs were responsible for over half the queries (300), in ProbeLog it takes almost a hundred smaller hubs.

## 5.6 ABLATION STUDIES

**How to select the probe distribution?** We showed (Sec. 5.2) that ProbeLog can generalize to real-world scenarios. Here, we conduct an ablation study, to test the effect of sampling probes from different distributions: (i) Dead-Leaves (Baradad Jurjo et al., 2021; Lee et al., 2001): a very coarse, hand-crafted generative model. (ii) ImageNet images. (iii) StableDiffusion (Rombach et al., 2022) samples using prompts of ImageNet-21K objects. (iv) COCO Images. Results, shown in Tab. 4, demonstrate a consistent pattern: probes sampled from distributions that are closer to the target concept obtain more accurate retrievals. However, we note that even quite different probe distribution can yield high retrieval accuracies. E.g., even though COCO images are typically of scenes rather than objects, they are effective probes, reaching a top-1 accuracy of more than $60\%$ when searching the INet-Hub by text. These results show that defining a general set of probes, which can retrieve a wide range of concepts is feasible. However, if there is access to probes from target concept's distribution, it is better to use them.

**How many probes are enough?** Fig. 5 presents the results of text retrieval on INet-Hub using increasing numbers of probes. More probes lead to better results but with diminishing gains. For example, $4,000$ COCO probes achieve good performance of $64.4\%$ top-1 accuracy, though it is possible to achieve a $68.2\%$ using $8,000$ probes. We then ablate how many probes should be taken into account from each logit $(r)$. Fig. 8 shows the top-1 accuracy on the INet-Hub against $r$, the number of top activating probes taken from each query. We can clearly see a peak when choosing $r = 50$, which aligns with our intuition: selecting too many probes also includes irrelevant ones, conversely, selecting too few probes adds too much variance to the estimate of the truncated distance.

## 6    DISCUSSION

**Non-random probe selection.**    We proposed an approach for searching models that can recognize a target concept. Our approach probes each model with $4,000$ COCO images to produce the representation of each logit. However, we believe this number can be reduced substantially. For instance, while we chose the set of probes at random, it is likely that a smaller and more curated of probes exists. Specifically, core-set methods, which aim to reduce the number of training data, could potentially reduce this number. Another direction is to use advanced collaborative filtering ideas which take into account the statistics of logit values. We believe this is a fruitful avenue for future research.

**Scaling-up to entire repositories.**    While our model zoos already have $1,500$ large models, including ViTs (Dosovitskiy, 2020) and RegNet-Ys Radosavovic et al. (2020), model repositories may contain millions of models. We tested our approach on smaller hubs mainly because we did not have the resources to probe and label a million models. However, this investment is feasible by industry standards: Using a single RTX-A5000 GPU we were able to probe each model with $4,000$ images in $\sim 12$ seconds on average. Meaning, probing a million classification models would require $\sim 3300$ GPU hours. After this one-time effort, probing new models uploaded to HuggingFace is relatively simple and requires just a single GPU dedicated for the task ($\sim 12M$ function evaluations a day). Having the representations for all models, the search is fast as our search algorithm operates in a space of a few tens of dimensions, where retrieval from even a billion entries is possible (Johnson et al., 2019; Jayaram Subramanya et al., 2019; Chen et al., 2021). Moreover, the descriptors are much lighter than actual model weights, and storing them is quite cheap. E.g., our INet-Hub model take $400GB$ of memory, but their logit descriptors for $8,000$ probes only consume $1.4GB$ of storage.

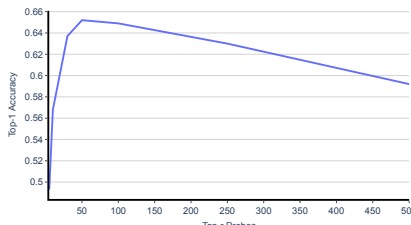

Figure 8: *Number of Considered Probes from each Logit.* We test different numbers of top-activating probes kept from each logit, for text-based logit retrieval. Taking $50$ probes performs best. Too many probes includes some that are irrelevant to the logit, while too few probes increases the variance of the truncated distance estimate.

## 7    LIMITATIONS

**Extension beyond classification models.**    Our proposed method embeds each logit of each model on its own. This will require modification for generative models where the output dimensions do not explicitly encode the learned concepts. While some works attempted to search for generative adapters (Lu et al., 2023), they typically required many more ($50,000$) probes as their descriptors summarize the distribution of probe outputs. We believe that our methodology, where the inputs are ordered and fixed for all models, can reduce the number of probes substantially.

**Out-of-distribution concepts.**    To enable search for diverse concepts we sampled probes from the COCO dataset (Lin et al., 2014) which does not contain just centered objects but also entire scenes. Still, these probes does not represent all concepts, e.g. medical concepts. Successfully finding far OOD concepts will require selecting a probe distribution that is better aligned to these concepts.

## 8    CONCLUSION

In this paper we propose an approach for searching for models in large repositories that can recognize a target concept. We first probe all models with a fixed, ordered set of probes, and define the values from each output dimension (logit) across all probes as a descriptor. Our proposed truncated euclidean distance score can classify logits within a closed set of concepts at similar or better accuracy as previous methods while being much simpler. By calibrating each logit according to its distance to the background concepts, we mitigate the hubness issue, allowing to search models by text. We evaluate our approach on real-world models, and show it generalizes well to in-the-wild models collected from HuggingFace. Our method retrieves models that are significantly more accurate than zero-shot CLIP, and ranks models according to accuracy on the query task.

## 9 REPRODUCIBILITY

We presented a simple approach for classification model search that can be readily reproduced. To facilitate future research, we provide a complete implementation of our method in the supplementary materials. Due to storage limitations, the trained models and probing datasets (requiring several hundred GB) cannot be included with this submission. However, we commit to releasing all datasets and trained models upon paper acceptance to ensure full reproducibility of our results.

## 10 USE OF LARGE LANGUAGE MODELS

In this paper we used Large Language Models (LLMs) for polishing the text. We prompted the LLM with small text segments (a few sentences at most) which we were not fully satisfied with and asked it to refine them. We emphasize that we manually observed every proposed modification, only accepting changes that improve the clarity of the text while preserving the original meaning.

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

## A  MODEL RETRIEVAL VIA IMAGE QUERIES

In many cases, describing an abstract visual concept using an example is much easier than describing the same concept by text. We therefore extend our search approach to a search-by-image setting. In this setting the query is a small set of images, and the expected response is a model logit which recognizes the main concept present in the query image set. This allows for users to search for complex concepts using known samples, rather than a long text description.

We adapt ProbeLog to accept such a small set of images as its input query. We denote CLIPs (Radford et al., 2021) image encoder as $\mathcal{I}_{clip}$ and the input set of query images as $q_1, ..., q_s$. We then probe CLIPs image encoder with each one of the input images $\mathcal{I}_{clip}(q_1), ..., \mathcal{I}_{clip}(q_s)$ and treat the average representation $c_q = \frac{1}{s}\Sigma_i \mathcal{I}_{clip}(q_i)$ as the representation of the main concept of the image set according to CLIP. Obtaining the similarities vector of the set with each probe is performed similarly to the text setting. Formally,

$$\phi_{query} = [< x_1, c_q >, ..., < x_n, c_q >] \tag{8}$$

Lastly, retrieval proceeds normally. We evaluate this approach on the INet-Hub where we have access to the distribution of images matching to each logit. I.e., we sample $k$ ImageNet (Deng et al., 2009) images from the query concept class, and use these as the query set for the retrieval. We compare ProbeLog to CLIP-Dissect (Oikarinen & Weng, 2023) and our normalized distance + anti-hub baseline. We experiment with a query image set sizes of $k = 1, k = 5$ and $k = 10$. Results are presented in Tab. 5.

Table 5: ***Search-by-Image Retrieval Results.*** We evaluate the Top-1 and Top-5 retrieval accuracies of searching by image queries, comparing our method and the baselines. We test query image set sizes of $k = 1, k = 5$ and $k = 10$. All methods use COCO images as probes. For a fair comparison, all experiments are performed with $4,000$ probes.

| Method | Top-1 Accuracy | | | Top-5 Accuracy | | |
|---|---|---|---|---|---|---|
| | $k = 1$ | $k = 5$ | $k = 10$ | $k = 1$ | $k = 5$ | $k = 10$ |
| CLIP-Dissect (WPMI) | 11.9% | 18.6% | 20.1% | 24.1% | 28.8% | 31.3% |
| CLIP-Dissect (SoftWPMI) | 9.4% | 15.2% | 16.0% | 18.5% | 28.1% | 29.0% |
| **ProbeLog (Ours)** | **23.5%** | **49.2%** | **55.3%** | **41.1%** | **70.4%** | **76.4%** |

## B  RETRIEVAL RANK VS. MODEL ACCURACY

We provide additional analysis, comparing the retrieved models accuracies vs CLIP's zero shot accuracy, at the model-level. For each query text, we retrieve the rank-k logit, and select the model it is a part of. We compare the accuracy of this model to CLIP zero-shot accuracy on the model's task and test set. The final number is averaged over all query texts. Results are presented in Tab. 6. It is clear that the retrieved models are substantially more accurate than zero-shot CLIP.

Table 6: ***Model Accuracy by Retrieval Rank.*** We compate ProbeLog retrieved models accuracy to CLIP zero-shot accuracy. It is clear that the top retrieved models significantly outperform CLIP.

| Rank | 1 | 2 | 3 | 5 | 7 | 10 | 15 | 20 |
|---|---|---|---|---|---|---|---|---|
| Retrieved Models | 92.7% | 92.5% | 92.3% | 91.7% | 91.6% | 90.8% | 89.8% | 89.4% |
| CLIP | 83.8% | 83.7% | 83.8% | 83.8% | 83.7% | 83.8% | 83.8% | 83.9% |

## C  BACKGROUND SET SIZE

As additional analysis, we wish to test the required background set size needed for ProbeLog. We therefore evaluated text-based search with background sets of different sizes, taken from ImageNet21k classes. For each size, we sampled $5$ non-overlapping sets. We then report the mean

and standard deviation for each size below. Our analysis (presented in Tab. 7) reveals that larger sizes do achieve better results but this saturates around 500 concepts. We note, that the one-time cost of computing 500 CLIP text encodings requires fewer resources than a single model probing. Therefore, using a background set of this size is highly practical.

Table 7: **_Top-1 Accuracy vs. Background Set Size._** ProbeLog's retrieval accuracy improves with larger background sets across both HuggingFace Hub and ImageNet Hub repositories.

| Background Set Size | HF-Hub Top-1 Acc. | INet-Hub Top-1 Acc. |
|:---:|:---:|:---:|
| 5 | 31.1±1.7% | 44.8±3.6% |
| 25 | 33.2±0.8% | 48.0±1.5% |
| 50 | 33.4±0.8% | 49.3±1.8% |
| 100 | 38.3±0.5% | 57.2±1.2% |
| 250 | 41.7±1.0% | 62.9±1.6% |
| 500 | 43.6±0.9% | 65.2±0.3% |
| 1000 | 44.3±1.0% | 66.9±0.5% |
| 2000 | 44.8±0.9% | 67.2±0.9% |

## D  PROBE SET AND MODELS TRAINING DATA

To further understand how to select the probing set, we conducted an additional experiment, where we tested if the similarity between the probe set and training set of the models affects the results. We tried 10 datasets as probe candidates, and computed the Frechet Distance (via DINO embeddings) from each one to ImageNet. We take ImageNet as the super-set of training data for our INet-Hub models. We then sampled probes from each of the 10 datasets, and computed text-based retrieval accuracy on the INet-Hub using them (averaged over 5 seeds). We present the results in Tab. 8. We highlight that the overall correlation of the DINO-FD and Top-1 accuracy is $-0.920$, showing that indeed the closer the probe dataset to the training data of the model, the better the retrieval.

Table 8: **_Performance vs. Dataset Domain Distance._** ProbeLog's retrieval accuracy increases as the Fréchet Distance from ImageNet decreases. This shows that smaller domain shift between the models training data and the probe set leads to better retrieval results.

| Dataset | ImageNet FD | Top-1 Acc. |
|:---:|:---:|:---:|
| ImageNet | 0.0 | 86.1% |
| COCO | 1347.7 | 64.4% |
| SD | 1500.9 | 73.9% |
| Food101 | 4960.7 | 22.3% |
| OxfordFlowers | 6712.1 | 15.7% |
| CIFAR100 | 7903.8 | 42.3% |
| StanfordCars | 7742.9 | 12.3% |
| GTRSB | 8437.1 | 2.9% |
| EuroSAT | 8813.3 | 5.7% |
| DeadLeaves | 10363.9 | 4.1% |

## E  INET-HUB DATASET DETAILS

To simulate a model hub with many classifiers, we train $1,500$ classifier models on different subsets of ImageNet classes. Each classifier is trained on a subset of between $15$ and $200$ classes, where the classes are chosen at random separately for each model. $90\%$ of the classifiers are initialized from a foundation model, and the rest $10\%$ are trained from scratch. The pre-training weights are selected from a set of $49$ different models spanning various architectures including ViTs (Dosovitskiy, 2020), ResNets (He et al., 2016), RegNet-Ys (Radosavovic et al., 2020), MLP Mixers (Tolstikhin et al., 2021), EfficientNets (Tan & Le, 2019), ConvNexts (Liu et al., 2022) and more. Each model is then trained for $2-5$ epochs. This process results in a model hub with over $85,000$ different logits to

search for and 1, 000 different fine-grained concepts. Below we list the possible pre-training weights of each model. All pre-training weights are taken from the timm library (Wightman, 2019).

- vit_base_patch32_clip_quickgelu_224.laion400m_e32
- vit_base_patch32_clip_224.laion400m_e32
- vit_base_patch32_clip_224.laion2b
- vit_base_patch32_clip_224.datacompxl
- convnext_base.clip_laiona
- convnext_base.clip_laion2b
- vit_base_patch32_clip_quickgelu_224.metaclip_400m
- vit_base_patch32_clip_quickgelu_224.metaclip_2pt5b
- vit_base_patch32_clip_224.metaclip_400m
- vit_base_patch32_clip_224.metaclip_2pt5b
- vit_base_patch32_clip_224.openai
- seresnextaa101d_32x8d.sw_in12k
- resmlp_24_224.fb_dino
- resmlp_12_224.fb_dino
- mixer_l16_224.goog_in21k
- mixer_b16_224.miil_in21k
- mixer_b16_224.goog_in21k
- resnetv2_152x2_bit.goog_in21k
- resnetv2_101x1_bit.goog_in21k
- resnetv2_50x1_bit.goog_in21k
- regnety_320.seer
- regnety_160.sw_in12k
- regnety_120.sw_in12k
- swin_tiny_patch4_window7_224.ms_in22k
- swin_base_patch4_window7_224.ms_in22k
- convnext_small.in12k
- convnext_tiny.in12k
- convnext_tiny.fb_in22k
- convnext_small.fb_in22k
- convnext_nano.in12k
- convnext_base.fb_in22k
- eca_nfnet_l0
- vit_base_patch16_224.dino
- vit_small_patch16_224.dino
- vit_base_patch16_224.mae
- vit_base_patch16_224.orig_in21k
- vit_base_patch32_224.orig_in21k
- vit_tiny_r_s16_p8_224.augreg_in21k
- vit_small_r26_s32_224.augreg_in21k
- vit_tiny_patch16_224.augreg_in21k
- vit_small_patch32_224.augreg_in21k

- vit_small_patch16_224.augreg_in21k
- vit_base_patch32_224.augreg_in21k
- vit_base_patch16_224_miil.in21k
- vit_base_patch16_224.augreg_in21k
- tf_efficientnetv2_s.in21k
- tf_efficientnetv2_m.in21k
- tf_efficientnetv2_l.in21k
- tf_efficientnetv2_b3.in21k

## F    HF-Hub Dataset Details

In order to test our method on real-world data, we collected more than 250 classifiers uploaded by users to hugging face where overall there 1300 possible logits in the dataset. The models are trained on a diverse set of models, and class names are given by free text. Hence, class names may not align perfectly as each user spells concept a bit differently (e.g., "Apple" vs. "Apples'). Moreover, some classifiers have different levels of granularity, such as "Car" vs. a specific car model "Toyota". For evaluation purposes only, we created a label mapping where we manually annotated to which classes each logit can be mapped. We follow these rules to allow mappings between labels: (i) Different spelling map to each other. (ii) An object can be mapped to a specific type of it, e.g. "cat" -¿ "siamese cat". (iii) A specific type of object can be mapped to its super-class e.g. "siamese cat" -¿ "cat". (iv) object of the same level of granularity that share a super class cannot be mapped to each other. For example, a "Golden Retriever" is not a good match for a "Husky". Additionally, we created an additional mapping which matches each class to its corresponding ImageNet concept when available.

Saying that we highlight that this label mapping is only needed for numerical evaluations purposes and that our method is completely unsupervised. When trying to search for models in-the-wild models the label mapping is not necessary.

## G    More Retrieval Samples

We present here 20 more randomly sampled retrievals of ProbeLog. We also provide a zip file with 80 more random retrievals as well. We can see that most of ProbeLog's retrievals are near misses. E.g., the class "crt screen" was matched to "television", and the class "space bar" was matched to "computer keyboard". Moreover, we can see that many mistakes which may seem far away visually are actually semantically related. For instance, when querying for "proboscis monkey" the logit returned was of the class "banana". Additionally, the query "hourglass" returned "barometer" as "barometer" is visually similar to an analog clock. However, there is still room for improvement as some mistakes are completely wrong such as "tarantula" and "baseball player".

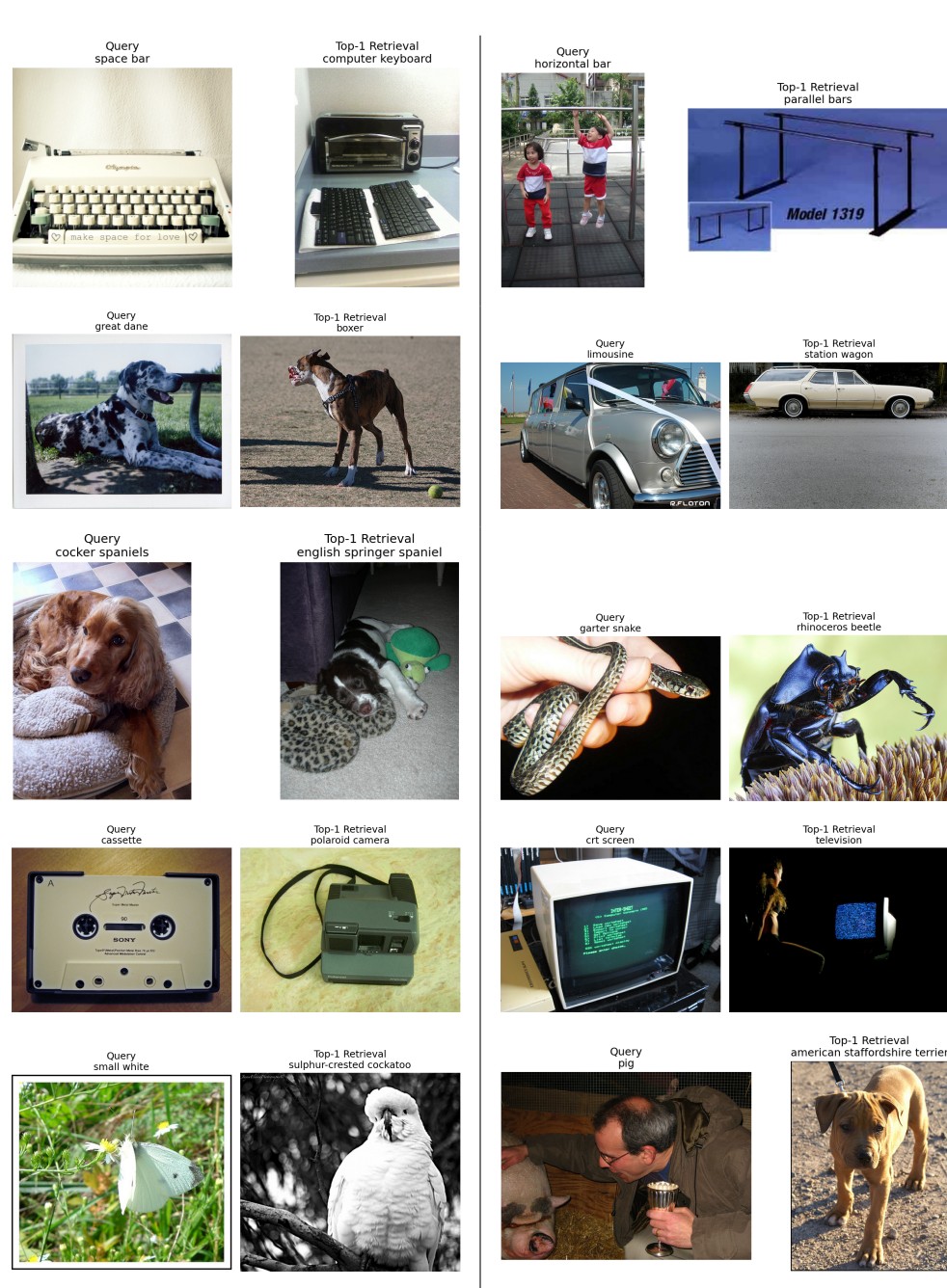

Figure 9: ***Retrieval Failure Cases.*** Visualization of random failure cases of ProbeLogs. Each pair shows a query image (left) and the top-1 retrieved result (right). These examples highlights that most of ProbeLogs mistakes are either semantically or visually similar classes.

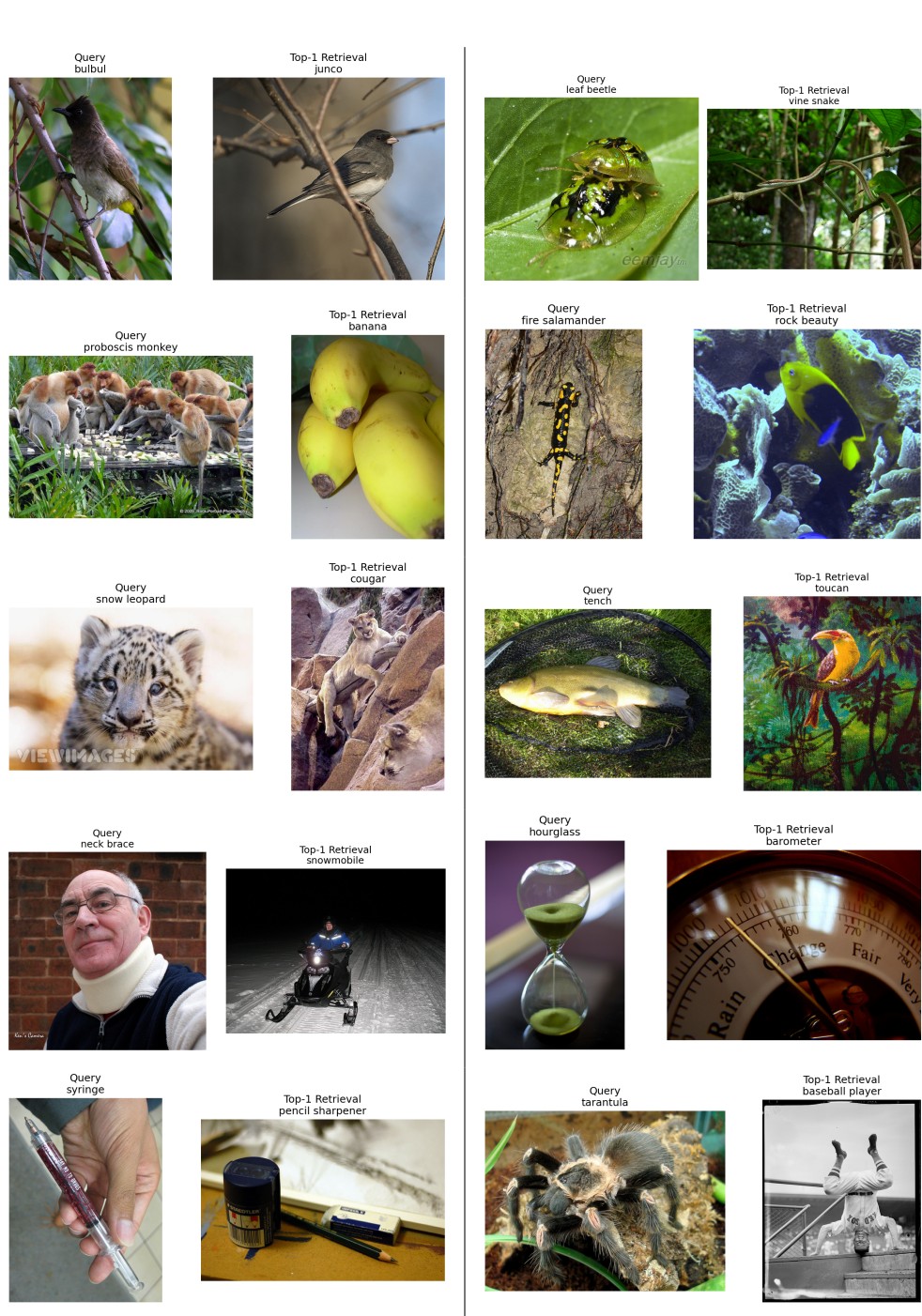

Figure 10: *Retrieval Failure Cases.* Visualization of random failure cases of ProbeLogs. Each pair shows a query image (left) and the top-1 retrieved result (right). These examples highlights that most of ProbeLogs mistakes are either semantically or visually similar classes.

