# OpenReview forum: "Zero-Shot Model Search via Text-to-Logit Matching"
_ICLR.cc/2026/Conference — ICLR 2026 Conference Withdrawn Submission_

### Official Review · Reviewer_zazE · 2025-10-28

**Soundness:** 1
**Presentation:** 2
**Contribution:** 1
**Rating:** 2
**Confidence:** 4

**Summary:**

The paper considers zero-shot model search with a logit-matching based method. The authors argue that many models in repositories such as Hugging Face lack textual metadata, making conventional text-based search insufficient. ProbeLog aims to address this by probing models with a fixed set of images, constructing descriptors for each logit, and comparing them with CLIP-derived concept descriptors. However,retrieving models that lack textual documentation is uncommon and not particularly useful in practice. Moreover, the method involves straightforward application of existing techniques (probing, CLIP similarity, and distance calibration) with no significant theoretical or algorithmic innovation.

**Strengths:**

- The paper tries to solve a fairly new problem that is retrieving models in large model repositories without textual information.
- The paper creates two model zoos for this task
- The writing is clear.

**Weaknesses:**

- The motivation for the problem itself is weak. In practice, models that lack texts in both metadata and comment section are often incomplete, unverified, or low-quality. Such models are rarely useful for end users and typically should not appear in search results. Consequently, the proposed setting addresses an unrealistic scenario with limited practical or research value.

- The paper argues that “most models are undocumented” based on counting Hugging Face model cards, but this ignores that most undocumented models are toy experiments, checkpoints, or duplicates. The paper does not demonstrate use cases that users actually need to retrieve such models.

- The method essentially reuses existing CLIP embeddings and probing techniques with minor heuristic modifications. There is little conceptual innovation or theoretical insight.

**Questions:**

- what are the concrete use cases for this method? how often do user want to retrieve models without any textual metadata?
- what are the performance of the retrieved models? although the retrieval accuracy is high, do the retrieved model actually help the users with their tasks, given that most undocumented models are of low quality?

---

> ### Author Response · Authors · 2025-12-01
>
> We thank the reviewer for reviewing our paper.
>
> **Task & motivation.** We respectfully disagree that retrieving undocumented models is of limited practical value. While many undocumented models may indeed be low-quality or intermediate checkpoints, we believe that some are useful, well-performing classifiers that simply lack proper documentation. Without a weight-based search method, such models are difficult to find and remain inaccessible to users who rely solely on text-based search.
>
> **What is the performance of the retrieved models?** In Tab. 3 we compared the retrieved models performance to CLIP zero-shot binary classification. ProbeLog’s top retrievals achieve substantially higher PR-AUC than both CLIP zero-shot classification and the average model in the repository, indicating that the top retrieved models are indeed accurate.

---

### Official Review · Reviewer_SCNZ · 2025-10-31

**Soundness:** 2
**Presentation:** 2
**Contribution:** 2
**Rating:** 4
**Confidence:** 3

**Summary:**

Model zoos like Hugginface contain a ton of models and a sizable portion (~60%) of them are not properly documented or annotated. This paper tackles the problem of searching for relevant classification model for a textual given concept e.g., dog or panda.

- The proposed approach (ProbeLog) works in zero-shot setting.
- It computes Euclidean distance between probe responses for logits and textual descriptions.
- The paper proposes two new datasets (INet-Hub and HF-Hub) for the setup.
- The proposed method yields superior results as compared to baseline.

**Strengths:**

1. The proposed method outperforms the baseline CLIP-Dissect on Top-1 and Top-5 retrieval accuracies for text-based retrievals and logit classification.
2. The paper targets the hubbness problem faced by earlier approaches. In particular, they choose 500 classes from ImageNet-21K to calibrate the distances of each logit. Table 4 presents an ablation study over othe datasets.
3. Ablations studies are perfomed over: (i) Selecting probes: access to probes from target concept’s distribution is useful; (ii) Number of probes to select: ~4000 probes provides a balance between accuracy and efficiency.
4. The paper presents results with a 1500 models. However, it presents a compelling case of scaling to millions of models. For instance, a one time investment of ∼3300 GPU hours (RTX-A5000 GPU) would be required to proble million classification models. Thereafter, a single dedicated GPU would do the job.

**Weaknesses:**

1. Section 4 could be better written.

(i) Please define $n$ explicitly (assumed it to be number of probes).
(ii) Better notation could be used for cosine similarity. $CLIP(x_1, c)$ can be confusing.
(iii) Creation of our zero-shot text-based logit descriptors should be properly discussed in the text as well.

2. The paper should include more baselines since the datasets are proposed in the paper itself. How about comparing against the model retrieved by existing hugginface search or a web search query?
3. The paper seem to be proposed for a single concept (e.g., dog, pandas) at a time. However, real usage is typically about a specific domain (say animal classification) rather than a single concept. Can the propsal be extended to union or intersaction or multiple concepts? or can it work out of the box?

**Questions:**

1. Can the model be extended to say text classification methods?
2. Have you experimented with methods other than CLIP? How do the recent VLM methods perform in this domain?
3. Please fix small things like: $j^th$ and $I.e.$.
4. What do you mean by "We then train the model on the selected data" (in #312)?

---

> ### Author Response · Authors · 2025-12-01
>
> We thank the reviewer for the detailed feedback and for highlighting several strengths of our work.
>
> **Sec. 4 writing.** We thank the reviewer for pointing these issues out and will clarify them for the next version of our paper.
>
> **Multiple concepts search.** While our experiments focus on retrieving models for a single target concept, the approach can be naturally extended to multiple concepts. One straightforward strategy is to run the search iteratively over each concept and retain only models that satisfy all of the queried concepts.
>
> **Extension to text classification.** Our method is formulated for classifiers, and in principle can be adapted to text classification by replacing CLIP with an appropriate text encoder and using textual probes instead of images. However, since text classifiers tend to cover a narrower range of concepts compared to image classifiers, we chose to focus on image models where the diversity of concepts provides a more comprehensive evaluation.

---

### Official Review · Reviewer_2d7A · 2025-10-31

**Soundness:** 3
**Presentation:** 2
**Contribution:** 2
**Rating:** 6
**Confidence:** 3

**Summary:**

This paper presents a method for searching for classification models within large model repositories. To be the best of my knowledge, the proposed application is novel and interesting. The proposed solution, although simple, seems to outperform existing alternatives that were not devised for this application in particular. The similarities between CLIP scores and classifier responses are used to assess the classifier's attunement with the query concept.

**Strengths:**

- The application is useful in practice and removes the burden of having highly technical knowledge to select classifiers from a repository.
- The proposed solution is simple and yields reasonable results, although far from optimal.
- There is value in the datasets, although it is not clear from the description how accurate the labeling is.

**Weaknesses:**

- Conceptually, we want the retrieved classifier(s) to be discriminative. It seems like it would be interesting to select logits that report high responses for high CLIP scores and low responses for low CLIP scores. However, this does not seem considered or even discussed by the authors. Clearly stating the difference between the proposed appication os model selection needs to be addressed.
- I would appreciate a discussion of the model search application within the context of model selection. This traditional topic in machine learning is very related to the proposed model search. How are the two different? What learnings can be used from model selection for model search?
- Why do the de-meaning and normalization in Equation (5) based on the mean and standard deviation of the entire logit? Why these statistics in particular? A detailed discussion leading to this choice would strengthen the work. Otherwise, it appears to be an ad hoc solution.
- What happens is the concept provided by the user is not well represented in the set of probes? For example, it would have been interesting to use a query concept and to remove the probes that actually belong to the query concept and see how performance varies. Ablation studies of this kind would add clarity for designing the probe set.
- Although the authors claim in Appendix F that the approach is unsupervised and the label mapping is only for numerical evaluations, the mapping may be used in the future to create new methods (and it surely was used by the authors to design their approach and select its hyperparameters). From that perspective, the approach is not unsupervised. It is like claiming that a classifier is unsupervised because one is providing a pre-trained version: one surely does not needs labels to use it in practice, but this fact does not make the classifier unsupervised.
- I think that the authors should add a section/paragraph on the ethical implications of their work. Automating model selection may produce biased or unfair results, in particular if the probe set has inherent biases for example. I'm not concerned by the choices of the authors but think a discussion might be useful for practitioners.
- How do the results change when using different (e.g., more recent) multimodal models?

**Questions:**

- How do the results change when using different (e.g., more recent) multimodal models?

---

> ### Author Response · Authors · 2025-12-01
>
> We thank the reviewer for the thoughtful comments and for noting that "the application is useful in practice".
>
> **Model search vs. model selection.** While traditional model selection assumes access to supervised samples from the target distribution, our setting is fundamentally different: users often do not have labeled examples, and the repository contains thousands of concepts that cannot all be annotated. For this reason, our method relies on comparing logit responses to CLIP-derived concept descriptors rather than using supervised validation data. This enables search even when no labeled samples of the desired concept are available.
>
> **Supervision.** As noted in Appendix F, label mappings are used strictly for evaluation, since many concepts in the repositories appear under slightly different names, e.g., “egg” vs. “eggs” or "car" vs. "automobile". Our method itself *does not use any labels*, is *training-free*, and *does not require a label mapping* for in-the-wild search. While we selected the number of top activating probes using these mappings, Fig. 8 shows that performance drop is minimal in the same range of values, indicating that the approach is not sensitive to this choice. We therefore insist that our method is unsupervised.

---

### Official Review · Reviewer_1n89 · 2025-11-01

**Soundness:** 3
**Presentation:** 3
**Contribution:** 2
**Rating:** 2
**Confidence:** 4

**Summary:**

The core problem this paper addresses is: how to find a model capable of recognizing a specific concept (e.g., "dog") within a massive (e.g., million-scale) model repository where documentation is severely lacking. The paper proposes a zero-shot model search method called ProbeLog, with the following technical process: The system first uses a fixed set of input images (called "probes") to "probe" each classification model in the repository. For each output dimension (logit) of a model, its responses (activation values) to all probes are recorded to form a vector. By calculating the distance between a text descriptor and all logit descriptors, the method can identify the logit (and its corresponding model) that best matches the text concept.

**Strengths:**

The online overhead is not that large.

The paper is well-written and flows smoothly.

**Weaknesses:**

The probe set is too large. When the model repository is very large, this method will also incur significant offline overhead.

The logit output is limited to classification models.

It should be compared with more methods in the field of model routing.

The method relies heavily on the samples in the probe set. If new images are significantly different (OOD) from the images in the probe set, the method is likely to fail.

**Questions:**

If time permits, could you add more experiments to explain why the performance is not good in specific domains?

---

> ### Author Response · Authors · 2025-12-01
>
> We thank the reviewer for reviewing our paper.
>
> **The probe set is too large.** In our experiments we used 4000 probes, which only requires several forward passes on modern GPUs. Moreover, as shown in Sec. 5.6, adding more probes leads to diminishing gains, hence reducing the probe set size will decrease accuracy by only a small amount.
>
> **The method relies heavily on the samples in the probe set.** Our method works well even when using probes from the COCO dataset which are out of distribution to the concepts that we aim to detect (ImageNet classes). We mentioned in the paper the limitation of our method in the extreme case where the concepts are far out-of-distribution of the probes, e.g., when they are medical concepts. In this case, we would need to be a bit more careful with our probe selection, but this is not a common case.

---

### Note · Authors · 2025-12-01

**Comment:**

We thank all reviewers for their efforts and helpful comments. We have addressed the reviewers' concerns below, and will submit the next version of the paper to another venue.

**Withdrawal Confirmation:**

I have read and agree with the venue's withdrawal policy on behalf of myself and my co-authors.